# Physiological Responses of Lettuce (*Lactuca sativa* L.) to Soil Contamination with Pb

**Elena Ikkonen * and Natalia Kaznina**

Institute of Biology of the Karelian Research Centre, Russian Academy of Sciences,
185910 Petrozavodsk, Russia

* Correspondence: likkonen@gmail.com

**Abstract:** Contamination of agricultural soils with heavy metal leads to a decrease in the crop quality and yield, as well as increases in public health risks. In this study, we aimed to evaluate the impact of soil contamination with lead (Pb) on the growth, photosynthesis, respiration, and coupling between these physiological processes, as well as temporal dynamics of Pb uptake and accumulation by lettuce (*Lactuca sativa* L.) plants. For this 46-day pot experiment, $Pb(NO_3)_2$ was mixed with loamy Retisol soil with the rate of 0, 50, and 250 mg kg$^{-1}$. No significant differences in plant biomass accumulation were found between plants grown on Pb-free and Pb-rich soil, but root-weight ratio, root-to-shoot weight ratio, and leaf area were lower, and the number of leaves and leaf weight per unit area were significantly higher in plants grown on soil contaminated with Pb than in their counterparts grown on Pb-free soil. The concentration of Pb in plant root and shoot followed the increase in soil Pb, with Pb content in the roots being higher than in the shoots. Soil Pb decreased chlorophyll content, net $CO_2$ assimilation rate and photosynthetic light use efficiency, but caused an increase in the leaf respiration rate regardless of whether respiration was performed in the light or in darkness. Increased ratio of respiration to photosynthesis reflects the shift in the carbon balance of lettuce plants toward carbon losses under stress conditions of soil contamination with Pb.

**Keywords:** lead; photosynthesis; respiration; chlorophyll; plant biomass; metal content; accumulation

## 1. Introduction

A serious problem for agricultural ecosystems is their pollution with technogenic components, including heavy metals. Lead (Pb), originating from various sources, is one of the most common soil contaminants. It is known that Pb does not decompose and is not naturally removed from the soil [1], so it accumulates, especially in soils with a high ability to absorb and deposit this heavy metal [2]. The accumulation of lead in soils is also facilitated by the low mobility of its compounds under high soil pH [3]. Markus and McBratney [4] reviewed data collected on lead concentrations in soils and showed high variability, with the maximum amount of lead reaching tens of grams per kilogram of soil. For the agricultural soils of the Moscow region, as an example, the Pb concentration in 2005 was close to 100 mg kg$^{-1}$ [2].

Even at low concentrations, heavy metals, including Pb, may be associated with potential risks to the environment or human health [5,6]. Lead is not a plant nutrient, and a high Pb content negatively affects the morphological, physiological, and biochemical traits of plants, with the decreased crop production and yield [7–9]. The number of studies [9,10] reviewed the adverse effects of lead on plant physiological processes, such as decreased germination and growth, changed activity of a wide range of enzymes of different metabolic pathways, disturbed mineral nutrition, water balance and gene

expression, enhanced production of reactive oxygen species. The Pb-mediated decline in photosynthetic rate was found to be a result of chlorophyll degradation, $CO_2$ limitation, reduced electron transport, and low activity of Calvin cycle enzymes [10–15]. However, some studies have shown the possibility of an increased rate of photosynthesis in plants growing on soils contaminated with lead [16,17]. Yang et al. [16] showed the enchanted photosynthetic rate in *Davidia involucrate* grown on a soil with 200 mg kg$^{-1}$ of Pb. For *Brassica chinensis*, Fu and Wang [17] found that the net $CO_2$ assimilation rate and chlorophyll fluorescence parameters increased with increasing soil Pb concentration up to 600 mg kg$^{-1}$.

Whereas the effects of soil Pb on the photosynthetic process is well studied [7,9,10,13–15], much less is known about an impact of lead contamination on the respiration metabolism [18,19], as well as about the electron partitioning between the cytochrome and alternative respiratory pathways. Along with the negative effect of lead on plant cell respiration [20], the respiration-stimulating effect of Pb was found in Pb-treated leaves of some species [19] and isolated mitochondria [18]. It is assumed that Pb-mediated increase in respiratory activity is associated with the activation of malate dehydrogenase and the mitochondrial ATP production [19], but the role of alternative respiratory pathways in an increased respiration is not yet clear. The balance between respiration and photosynthesis controls plant growth and productivity [21]; however, less information is available about the modification of coupling between these physiological processes under soil Pb stress. Moreover, it is uncertain whether Pb affects the degree of inhibition of respiration by light and hence the difference in the ratio of respiration and photosynthesis in the light and in darkness.

The uptake and accumulation of lead by plants varied between plants, with lower Pb uptake by tolerant than sensitive plants [22]. Very rare plant species can accumulate significant amounts of lead without significant damage of physiological processes [23]. Lettuce *Lactuca sativa* L., as a crop species important to the human diet, is currently cultivated over the world. A decrease in the growth of roots and shoots, as well as an imbalance in nutrition and a change in the chlorophyll content, were revealed for lettuce plants irrigated with lead-contaminated water for 15 days [7], but no significant effect of lead on root and shoot dry weight accumulation in lettuce plants was found by Đurđević et al. [24] and Silva et al. [25]. The reason for these differences may be that the accumulation of Pb in lettuce is genotype-dependent, as was established by Zhang et al. [26]. Silva et al. [27] showed significant DNA damage when lettuce crops were irrigated with water containing Pb, although seed germination parameters and seedling growth were not affected. In lettuce plants, an increased content of soil Pb reduced the contents of leaf macro- and micronutrients, as well as the content of photosynthetic pigments [28]. Pb-mediated depression of photosynthetic activity was found for lettuce plants [25], but the response of respiration and the coupling between photosynthesis and respiration to soil Pb contamination are not well documented. This study aimed to evaluate the physiological responses of lettuce plants to soil contamination with Pb in order to better understand strategies used by plants to tolerate this stress.

## 2. Materials and Methods

### 2.1. Plant Materials and Pb Treatments

For this plot experiment, we used sandy loam soil collected from the ploughed topsoil layer from a field site in the northwest region of Russia (61.826573, 33.179712). The soil was not fertilized nor subjected to pesticide treatments for some years before soil collection. The collected soil had quite low water holding capacity, low humus content (0.5–2.5%), 5.46 pH, 0.39% of total N, 1200 and 145 mg kg$^{-1}$ of available P and K, respectively. The available Pb content of soil under the study was $0.13 \pm 0.01$ mg kg$^{-1}$. The soil was air-dried and sieved with a 2 mm sieve. The entire volume of the dry soil was divided into parts and carefully mixed with Pb(NO$_3$)$_2$ with the rate of 0, 50, and 250 mg kg$^{-1}$

(0 Pb, 50 Pb and 250 Pb treatments, respectively). Before seed sowing, the soil substrates of each treatment were incubated under 21–23 °C of air temperature and 70–80% of the maximum soil water holding capacity for 14 days, and then parked into plastic 0.80 L pots with a soil bulk density of approximately 1.4 g cm$^{-3}$. Each of the 0 Pb, 50 Pb and 250 Pb treatments included eight pots.

The seeds of lettuce (*Lactuca sativa* L., var. Medvezhje ushko) were sown with six seeds per each pot. This variety is popular in the northwest region of Russia because of this high productivity and good nutritional qualities. The plant cultivation was conducted under controlled conditions of 16-h photoperiod, 23/20 °C day/night temperature, 250 µmol m$^{-2}$ s$^{-1}$ of photosynthetic photon flux density (PPFD). All pots were watered with distilled water every two days. One week after sowing, the seedlings were thinned to three per pot.

### 2.2. Chlorophyll Fluorescence and Chlorophyll Content

Maximum photochemical quantum efficiency of PSII ($F_v/F_m$) was measured using MINI-PAM (Walz, Effeltrich, Germany). Before measuring minimum and maximum fluorescence ($F_o$ and $F_m$, respectively) of leaves, they were dark-adapted for 30 min using leaf clips. The $F_o$ and $F_m$ values were taken by illumination with saturated light and used to calculate $F_v:F_m = (F_m - F_o)/F_m$. The chlorophyll content was measured on the same leaves as the chlorophyll fluorescence parameters using a SPAD-502 chlorophyll meter (Minolta, Tokyo, Japan). Among all treatments, the chlorophyll fluorescence and chlorophyll content parameters were measured on the 17th, 21st, 34th, and 46th days after plant sowing.

### 2.3. CO$_2$ Gas Exchange

Light response curves of CO$_2$ gas exchange were measured on the fully expanded leaves of 45-day seedlings using a portable photosynthesis system (HCM-1000, Walz, Effeltrich, Germany) at the leaf chamber temperature of 20 °C, relative air humidity of 60–70%, and 400 ± 20 ppm of CO$_2$. The leaf area-based net CO$_2$ exchange, stomata conductance ($g_s$), transpiration (Tr) rate, the ratio of intercellular ($C_i$) to ambient ($C_a$) CO$_2$ concentration were determined starting at 300 µmol PPFD m$^{-2}$ s$^{-1}$ and then 2000, 1200, 1000, 800, 300, 60, 40, 20, and zero PPFD. Readings were taken after steady state rates of CO$_2$ exchange were reached. The rates of leaf respiration in the darkness ($R_{d[area]}$) were taken after 30 min of zero irradiance. The apparent quantum yield of photosynthesis ($\alpha$) was calculated as the slope of the net CO$_2$ assimilation rate ($A_{n[area]}$) versus irradiance of 20, 40 and 60 µmol m$^{-2}$ s$^{-1}$ according to Garmash and Golovko [29]. To estimate the rates of leaf respiration in the light ($R_{l[area]}$), the Kok [30] method was used. The degree of light inhibition of respiration was calculated as $1 - R_l/R_d$. The light compensation point (LCP) was found as PPFD, where $A_n$ is equal to zero. The rate of photosynthetic electron transport ($J$), carboxylase ($v_c$) and oxygenase ($v_o$) activity of Rubisco under saturating light (1200 µmol PPFD m$^{-2}$ s$^{-1}$) were calculated according to Farquhar and Von Caemmerer [31]. Photosynthetic water use efficiency (PWUE) was defined as the ratio of the $A_n$ rate at 1200 µmol PPFD m$^{-2}$ s$^{-1}$ to the Tr rate. The gross photosynthesis rate ($A_g$) was determined as $A_n$ plus $R_l$. The ratios of $R$ to $A_g$ were calculated for the $A_g$ rates at 1200 µmol PPFD m$^{-2}$ s$^{-1}$ and designated as $R_l:A_n$ and $R_d:A_n$. Leaf mass per area (LMA) was calculated as the dry mass of leaf segment per segment area (2.7 cm$^2$) with the drying of the segment at 70 °C to a constant weight. The LMA values were used to calculate $A_n$ at 1200 µmol PPFD m$^{-2}$ s$^{-1}$, $R_l$ and $R_d$ per unit leaf mass unit ($A_{n[mass]}$, $R_{l[mass]}$ and $R_{d[mass]}$, respectively).

### 2.4. Leaf Relative Water Content

For measuring relative water content (RWC), leaf discs were collected from randomly selected leaves of each treatment. First, the fresh weight (FW) of the leaf disks was determined, and then the disks were placed in distilled water for rehydration. After disc incubation at 4 °C for 24 h, leaf turgid weight (TW) was determined. Then, leaf disks

were dried at 70 °C and the dry weight (DW) was determined. The RWC value was calculated as (FW−DW)/(TW−DW) × 100% according to González and González-Vilar [32].

### 2.5. Mitochondrial Respiratory Pathway

The Clark electrode (Oxygraph System Plus, Hansatech, UK) was used for an analysis of mitochondrial leaf respiration rates. Then, 30 mM of salicylhydroxamic acid (SHAM) were added to the 2 mL of 100 mM HEPES buffer solution (pH 7.5) as an inhibitor of alternative respiratory partway. Total respiration ($V_t$) in the absence of SHAM and SHAM-resistant ($V_{SHAM-res}$) respiration rate in the presence of SHAM was measured as described earlier [33]. The MLW liquid thermostat (VEB MLW PRUFGERATE-WERK, GDR) was used to keep a buffer solution temperature at a 23 °C level.

### 2.6. Plant Growth Parameters

For each treatment, five 17-, 21-, 34-, and 46-day-old lettuce seedlings were harvested, separated into shoots and roots. The total leaf number per plant was counted, and the total leaf area per plant was determined by leaf scanning and using the program "AreaS". Harvested seedlings shoots and roots were dried at 70 °C to constant weight and weighed. The shoot weight ratio (SWR) and root weight ratio (RWR) were calculated as the ratio between organ weight and total biomass. The root–shoot ratio (RSR) was calculated as the ratio between root weight and shoot biomass.

### 2.7. Leaf and Root Pb Content

Homogenized shoot and root samples of 0.2–0.3 g were digested with $HNO_3$ and HCl (Vekton, Saint-Petersburg, Russia). The concentration of Pb was determined by spectrophotometric atomic absorption (Shimadzu AA-7000, Kyoto, Japan) in the Core Facility "Analytical laboratory" of the Forest institute of KRC of RAS. The accumulation of Pb in shoot and root per seedling was calculated using data of organ Pb content and dry organ biomass.

### 2.8. Statistical Analysis

One-way ANOVA was used to evaluate the significant differences between the means with LSD test at the $p < 0.05$ level (Statistica software, v. 8.0.550.0, StatSoft, Inc., Tulsa, OK, USA).

## 3. Results

### 3.1. Plant Growth

Among the studied Pb content, soil contamination with lead did not have a significant effect on dry biomass accumulation in lettuce, but for 46-day-old plants, the leaf number was significantly lower, and the leaf area was higher in 0 Pb, than 50 Pb or 250 Pb plants (Figure 1a–e). No significant differences in the mean values of SWR, RWR, and RSR were found between 50 Pb and 250 Pb plants, regardless of the number of days after sowing (Figure 1f–h). However, for 34- and 46-day-old plants, the RWR were significantly lower in 50 Pb and 250 Pb than in 0 Pb plants, and RSW values were highest in 0 Pb plants 46 days after sowing. Soil contamination with Pb caused a significant increase in leaf mass per unit area (LMA), regardless of lead content in the soil (Table 1). The decrease in leaf area of 50 Pb and 250 Pb plants was accompanied by the increase in shoot weight, but no significant differences in shoot biomass were found between 0 Pb plants with the larger leaves and 50 Pb or 250 Pb plants with the smaller ones. The increase in the weight of a unit area of the leaves grown on the soil with Pb is reflected in increased LMA values.

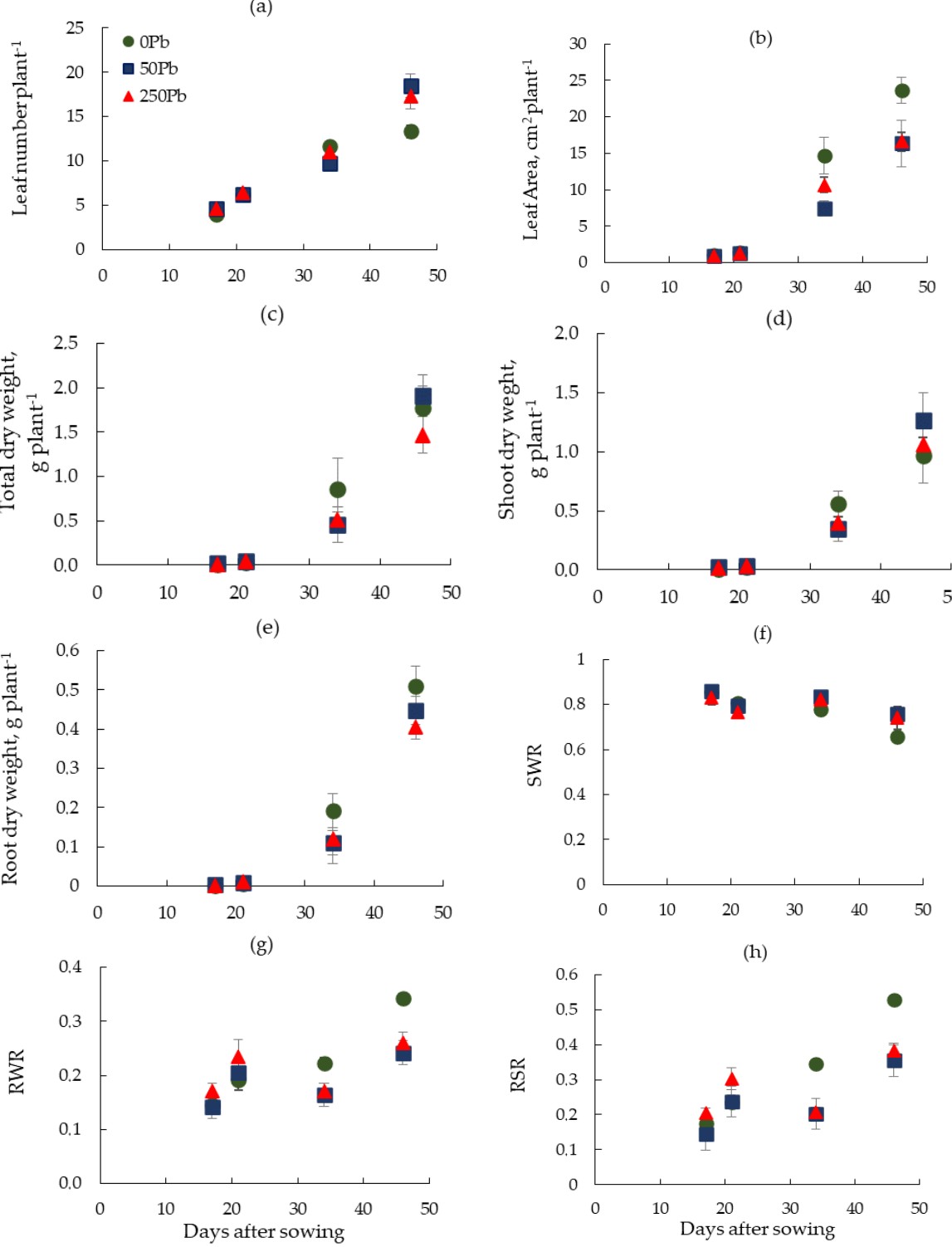

**Figure 1.** Leaf number (**a**), leaf area (**b**), total plant (**c**), shoot (**d**), root (**e**) dry weight, shoot-to-total-weight ratio (SWR, (**f**)), root-to-total-weight ratio (RWR, (**g**)), and root-to-shoot-weight ratio (RSR, (**h**)) of *L. sativa* grown on the soil with Pb content of 0 (0 Pb), 50 (50 Pb), and 250 (250 Pb) mg[Pb(NO$_3$)$_2$] kg$^{-1}$.

**Table 1.** Physiological parameters of *L. sativa* grown on the soil with Pb content of 0 (0 Pb), 50 (50 Pb), and 250 (250 Pb) mg[Pb(NO$_3$)$_2$] kg$^{-1}$.

| Parameter | 0 Pb | 50 Pb | 250 Pb | $p$ |
|---|---|---|---|---|
| LMA, g m$^{-2}$ | 19.3 ± 0.8 b | 23.5 ± 0.6 a | 22.9 ± 0.7 a | <0.05 * |
| Tr, mmol m$^{-2}$ s$^{-1}$ | 1.06 ± 0.09 a | 0.93 ± 0.10 a | 0.94 ± 0.13 a | 0.631 ns |
| $q_s$, mmol m$^{-2}$ s$^{-1}$ | 81 ± 9 a | 69 ± 7 a | 89 ± 10 a | 0.532 ns |
| $A_{n1200[area]}$, µmol m$^{-2}$ s$^{-1}$ | 7.1 ± 0.6 a | 5.8 ± 0.6 ab | 4.9 ± 0.9 b | <0.05 * |
| $A_{n1200[mass]}$, nmol g$^{-1}$DW s$^{-1}$ | 367 ± 28 a | 245 ± 26 b | 231 ± 41 b | <0.05 * |
| $J$, µmol m$^{-2}$ s$^{-1}$ | 47 ± 4 a | 46 ± 4 a | 43 ± 6 a | 0.738 ns |
| $v_o$, µmol m$^{-2}$ s$^{-1}$ | 1.9 ± 0.3 a | 2.1 ± 0.1 a | 1.9 ± 0.3 a | 0.797 ns |
| $v_c$, µmol m$^{-2}$ s$^{-1}$ | 8.9 ± 0.6 a | 8.7 ± 0.7 a | 8.1 ± 1.0 a | 0.744 ns |
| $C_i$:$C_a$ | 0.59 ± 0.04 a | 0.61 ± 0.02 a | 0.62 ± 0.07 a | 0.807 ns |
| PWUE, µmolCO$_2$ mmol$^{-1}$H$_2$O | 6.8 ± 0.7 a | 6.4 ± 0.6 a | 6.4 ± 0.9 a | 0.762 ns |
| $\alpha$, µmolCO$_2$ µmol$^{-1}$ quant | 0.018 ± 0.001 a | 0.016 ± 0.001 ab | 0.014 ± 0.002 b | <0.05 * |
| RWC, % | 71 ± 2 ab | 69 ± 2 b | 76 ± 1 a | <0.05 * |
| LCP, µmol m$^{-2}$ s$^{-1}$ | 33 ± 5 c | 54 ± 6 b | 78 ± 15 a | <0.01 ** |
| $R_{l[area]}$, µmol m$^{-2}$ s$^{-1}$ | 0.58 ± 0.08 b | 0.86 ± 0.08 ab | 1.00 ± 0.17 a | <0.01 ** |
| $R_{l[mass]}$, nmol g$^{-1}$DW s$^{-1}$ | 30 ± 4 b | 37 ± 3 ab | 43 ± 6 a | <0.01 ** |
| $R_{d[area]}$, µmol m$^{-2}$ s$^{-1}$ | 0.71 ± 0.07 b | 0.99 ± 0.08 ab | 1.14 ± 0.18 a | <0.01 ** |
| $R_{d[mass]}$, nmol g$^{-1}$DW s$^{-1}$ | 37 ± 4 b | 42 ± 3 ab | 50 ± 4 a | <0.01 ** |
| $R_l$:$R_d$ | 0.81 ± 0.04 a | 0.87 ± 0.014 a | 0.87 ± 0.015 a | 0.959 ns |
| $A_{g[area]}$, µmol m$^{-2}$ s$^{-1}$ | 7.6 ± 0.5 a | 6.6 ± 0.5 ab | 5.9 ± 0.8 b | <0.05 * |
| $R_l$:$A_g$ | 0.07 ± 0.01 c | 0.14 ± 0.02 b | 0.20 ± 0.05 a | <0.001 *** |
| $R_d$:$A_g$ | 0.09 ± 0.01 c | 0.16 ± 0.02 b | 0.22 ± 0.06 a | <0.001 *** |
| $V_t$, µmolO$_2$ g$^{-1}$DW h$^{-1}$ | 100 ± 15 a | 106 ± 9 a | 96 ± 7 a | 0.987 ns |
| $V_{SHAM-res}$, µmolO$_2$ g$^{-1}$DW h$^{-1}$ | 45 ± 9 a | 45 ± 7 a | 47 ± 4 a | 0.998 ns |
| $V_{SHAM-sen}$:$V_t$ | 51 ± 7 a | 55 ± 9 a | 51 ± 3 a | 0.968 ns |

LMA, leaf mass per area; Tr, transpiration rate; gs, stomatal conductance; $A_{n1200[area]}$ ($A_{n1200[mass]}$), area-based (mass-based) net CO$_2$ assimilation rate; $J$, electron transport rate; $v_o$, oxygenase activity of Rubisco; $v_c$, carboxylase activity of Rubisco; $C_i$:$C_a$, the ratio of intercellular to ambient CO$_2$ concentration; PWUE, photosynthetic water use efficiency; $\alpha$, apparent quantum yield of photosynthesis; RWC, relative water content; LCP, light compensation point; $R_{l[area]}$ ($R_{l[mass]}$), area-based (mass-based) respiration in the light; $R_{d[area]}$ ($R_{d[mass]}$), area (mass) area-based (mass-based) respiration in the darkness; $R_l$:$R_d$, the ratio of $R_l$ to $R_d$; $A_g$, gross CO$_2$ assimilation rate ($A_g = A_n + R_l$); $R_l$:$A_g$, the ratio of $R_l$ to $A_g$; $R_d$:$A_g$, the ratio of $R_d$ to $A_g$; $V_t$, total respiration; $V_{SHAM-res}$, SHAM-resistant respiration; $V_{SHAM-sen}$:$V_t$, the ratio of SHAM-sensitive ($V_{SHAM-sen}$) respiration to $V_t$. Different letters indicate significant differences between the means at $p<0.05$. Asterisks denote significance levels: * $p<0.05$, ** $p<0.01$, *** $p<0.001$; ns, not significant.

### 3.2. Effect of Pb on Leaf Gas Exchange of Lettuce

Soil contamination with Pb in the concentrations under the study decreased the net CO$_2$ assimilation ($A_n$) rate of lettuce leaves regardless of the soil Pb content and light intensity at which gas exchange was determined (Figure 2). The area- and mass-based rates of net CO$_2$ assimilation under light saturation ($A_{n1200[area]}$ and $A_{n1200[mass]}$, accordingly) were, respectively, 31 and 37% lower in 250 Pb plants than in plants grown on Pb-free soil (Table 1). Also, the $F_v$:$F_m$ values, which may be used as a direct indicator of photosynthetic activity and plant resistance to stress [34], were significantly lower in the 250 Pb plants, than in their Pb-free grown counterparts (Figure 3a).

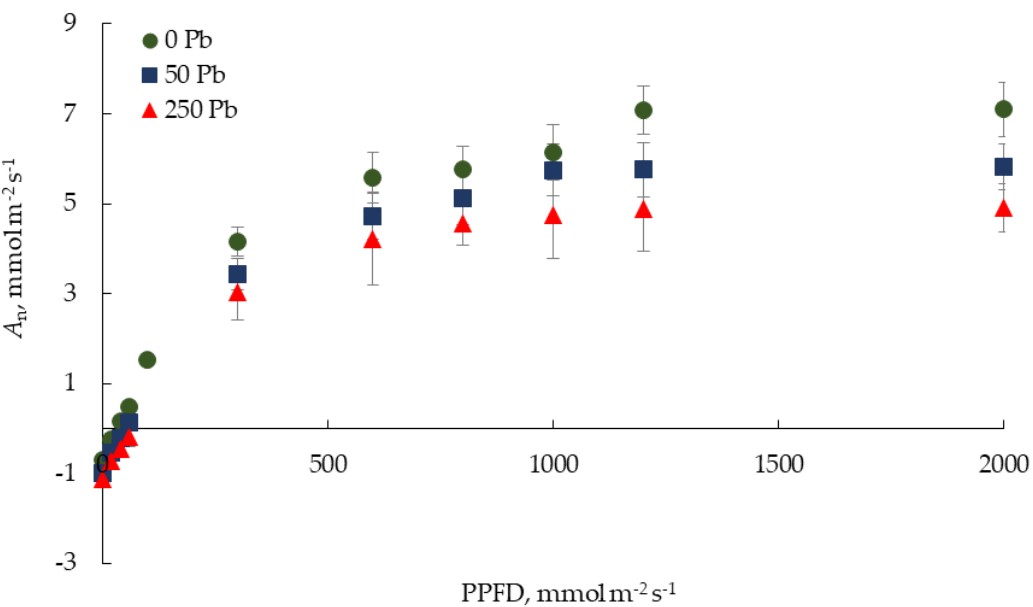

**Figure 2.** Light response of net assimilation rate ($A_n$) of *L. sativa* grown on the soil with Pb content of 0 (0 Pb), 50 (50 Pb), and 250 (250 Pb) mg[Pb(NO₃)₂] kg⁻¹.

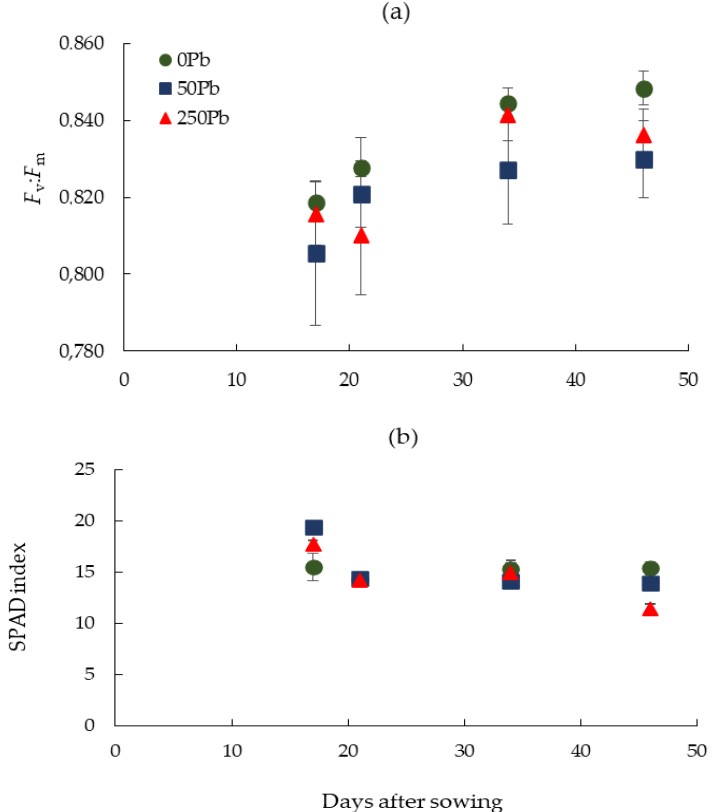

**Figure 3.** Temporal dynamics of the maximal quantum yield of PSII photochemistry ((**a**), $F_v{:}F_m$) and SPAD index (**b**) of *L. sativa* grown on the soil with Pb content of 0 (0 Pb), 50 (50 Pb), and 250 (250 Pb) mg[Pb(NO₃)₂] kg⁻¹.

The leaf respiration rate was significantly higher in the 250 Pb than 0 Pb plants, regardless of whether respiration occurred in the light or darkness, and regardless of whether the rate was calculated for the area or mass of the leaves ($R_{l[area]}$, $R_{l[mass]}$, $R_{d[area]}$, $R_{d[mass]}$, Table 1); however, this Pb effect was not confirmed by a polarographic study of $O_2$ uptake by leaves ($V_t$, Table 1). Furthermore, no effect of soil Pb on the partitioning between SHAM-resistant and SHAM-sensitive respiratory rates ($V_{SHAM-res}$, $V_{SHAM-sen}$:$V_t$, Table 1) was found in this study.

For the studied lettuce of all treatments, the degree of light inhibition of leaf respiration (1− $R_l$:$R_d$) varied, on average, between 13 and 19%. In accordance with the increase in Pb content in the soil, the ratio of leaf respiration to photosynthesis significantly increased, regardless of whether respiration was carried out in the light or darkness ($R_l$:$A_g$, $R_d$:$A_g$, Table 1).

### 3.3. Pb Content and Accumulation in Shoot and Root of Lettuce

At the end of the experiment, the concentration of available Pb in the soil was 0.10 ± 0.02, 4.03 ± 0.50, and 41.6 ± 3.6 mg kg$^{-1}$, respectively, for 0 Pb, 50 Pb, and 250 Pb treatment.

The Pb content was significantly higher in roots, than shoots of lettuce plants, regardless of the Pb concentration in the soil (Figure 4). For both shoot and root, the Pb content increased following the increase in soil Pb content. In contrast to the roots, for which no significant difference in lead content between 21-, 34-, and 46-day-old plants was found (Figure 4b), the lead content in the shoots of 50 Pb and 250 Pb plants increased with plant growth (Figure 4a).

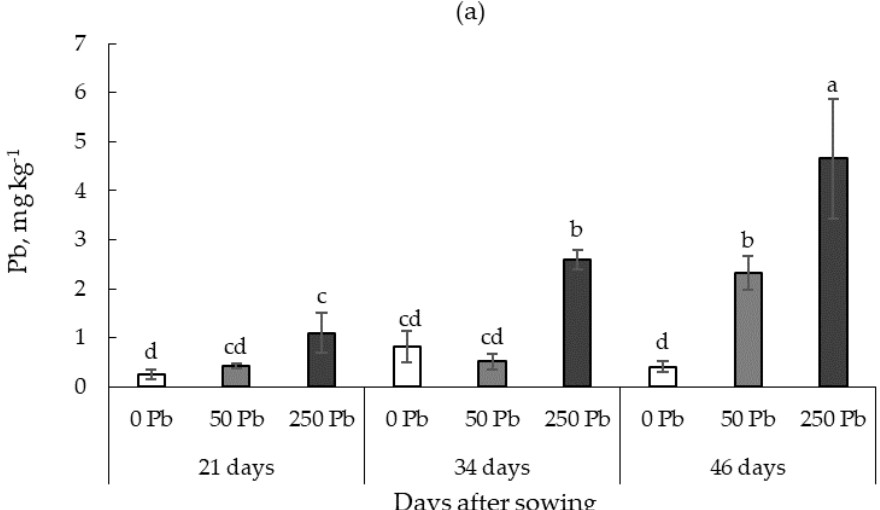

(a)

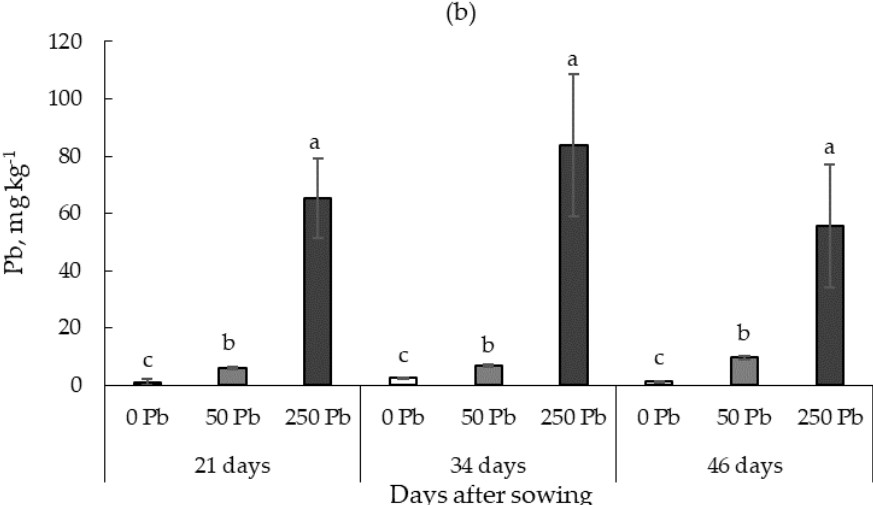

**Figure 4.** The content of Pb in the shoot (**a**) and root (**b**) of *L. sativa* grown on the soil with Pb content of 0 (0 Pb), 50 (50 Pb), and 250 (250 Pb) mg[Pb(NO₃)₂] kg⁻¹ at the 21st, 34th and 46th days after sowing. Different letters indicate significant differences between means at the $p < 0.05$.

While lead accumulation in the shoot and root was negligible for plants grown on Pb-free soil, it was significant in 50 Pb and especially in 250 Pb plants (Figure 5). In these plants, the Pb accumulation followed the accumulation of dry biomass with more significant accumulation in the roots than shoots; even though the accumulation of biomass in the roots was significantly lower than in the shoots (Figure 1d,e).

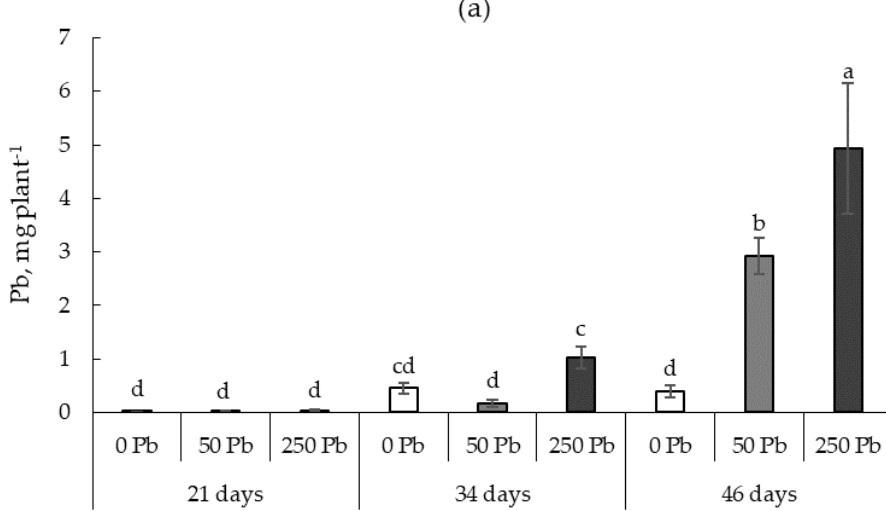

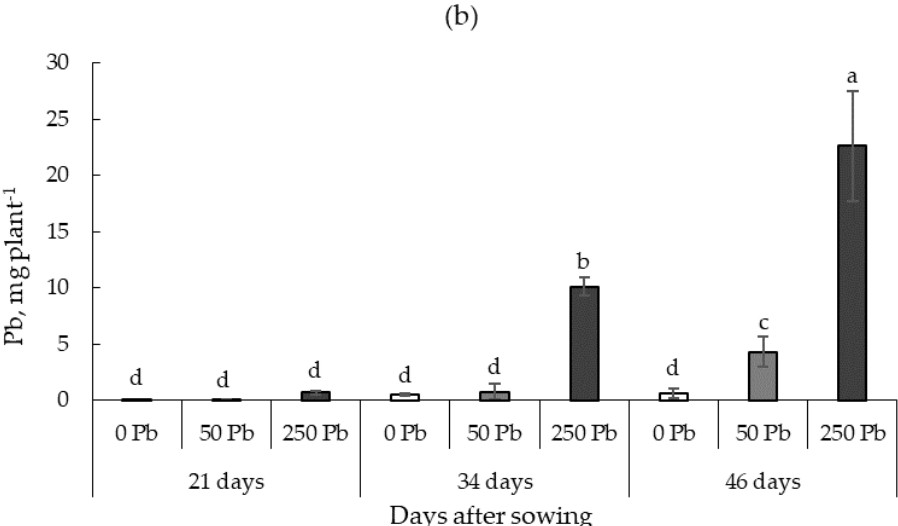

**Figure 5.** The accumulation of Pb in the shoot (**a**) and root (**b**) of *L. sativa* grown on the soil with Pb content of 0 (0 Pb), 50 (50 Pb), and 250 (250 Pb) mg[Pb(NO₃)₂] kg⁻¹ at the 21st, 34th and 46th days after sowing. Different letters indicate significant differences between means when $p < 0.05$.

## 4. Discussion

For a wide range of cultivated plants, soil contamination with Pb in high doses has been found to inhibit plant growth and biomass accumulation [7,35,36]. Low and moderate Pb stress studied in our work did not strongly affect total biomass accumulation by lettuce; however, it affected the allocation of biomass to organs. Đurdecic et al. [24] also did not reveal any significant effect of Pb on leaf and root dry mass accumulation for lettuce plants. One of the strategies for plant adaptation to changing environmental conditions can be a change in the accumulation of biomass in roots and shoots. Under stress conditions, including soil contamination with Pb, the biomass allocation strategy can optimize resource utilization to support optimal plant growth [37]. Our results revealed that Pb decreased root-to-total-plant-weight ratios (RWR), as well as root-to-shoot ratios (RSR) which can be caused by inhibition of root growth and elongation stronger than shoot growth and can be one of the plant mechanisms to maintain functional balance under soil Pb stress.

This study confirmed that Pb negatively affects photosynthesis, one of the main physiological processes of plants. Our finding that photosynthetic activity depressed under soil Pb contamination is according to the studies that have found that Pb can affect photosynthetic apparatus, including Calvin cycle, $CO_2$ ability, photosynthetic pigment biosynthesis [7,16,38,39]. However, Fu and Wang [17] showed opposite results for *Brassica chinensis* with the increase in the net photosynthetic rate and chlorophyll fluorescence parameters when soil Pb content was increased to 600 mg kg⁻¹. In our study, Pb soil contamination inhibited lettuce photosynthetic activity without any stimulatory effect of low Pb concentration on stomatal conductance, chlorophyll content, or net assimilation rate, as found in early studies [15–17].

The net $A_n$ rate, as an integral indicator of photosynthetic apparatus, is determined by the relative activity of a complex of reactions and processes. Because no significant effect of soil Pb was found for $g_s$ and $C_i$:$C_a$ ratio (Table 1), lead-mediated decrease in the $A_n$ rate was not connected with stomatal closure and $CO_2$ limitation of photosynthesis. These results confirm the findings of Silva et al. [25], such that Pb-mediated limitations of photosynthesis occurred from reasons other than reduced $CO_2$ availability. It might be expected that the decrease in $A_n$ would be associated with the inhibition of Rubisco activity and a decrease of electron transport rate, but this is, however, not the case, because this study did not reveal the effect of lead on such photosynthesis parameters as $J$, $v_o$ and

$v_c$ (Table 1). The depression of the $A_n$ rates in lettuce grown on Pb-rich soil related to the decrease of apparent quantum yield of photosynthesis ($\alpha$, Table 1), which is defined as the moles of $CO_2$ fixed per mole of quanta absorbed and reflected the efficiency with, which light, is converted into fixed carbon. This parameter is not a constant and varies depending on the conditions under which photosynthesis occurs [29,40,41]. One of the factors responsible for the change in the $\alpha$ value can be some adjustments in the pigment complex [29], which can be considered as an acclimation process to changes in plant growth conditions [42,43]. This study revealed a decrease in the chlorophyll content in the leaves of 46-day-old plants grown on the soil with $Pb(NO_3)_2$ content of 250 mg $kg^{-1}$ compared to 0 Pb plants (Figure 3b), which could lead to a decrease in the $\alpha$ and $A_n$ values. Ahmed et al. [44] showed that the total chlorophyll content of lettuce plants is sensitive to metals content in soil solution. The results of our study are consistent with the findings demonstrating decreased chlorophyll content in plants grown on soils contaminated with lead [15,20]. The Pb-mediated changes in the pigment complex can be caused by the destruction of the chloroplast ultrastructure [45], decreased photocatalytic activity chlorophyll [20], decreased chl*a*/chl*b* ratio [7], although the carotenoid content was found to be independent of lead [46].

In contrast to light use efficiency for photosynthesis, this study found no statistical evidence that soil Pb affects photosynthetic water use efficiency (PWUE, Table 1). At the same time, the relative water content in the leaves of 250 Pb plants was statistically higher than in the leaves of 0 Pb and 50 Pb ones (RWC, Table 1). Such nutrients, like K, P, Mg, Mn, Zn and Si, can affect plant processes related to water use efficiency and improve plant PWUE status [38,47–49]; therefore, it can be indirectly assumed that lead did not affect the uptake of these elements by lettuce under the study.

Our results showed that leaf respiration is enhanced in lettuce grown in lead-contaminated soil, consistent with the studies showing a similar response of leaf and mitochondrial respiration to lead [18,19]. The Pb-mediated increase in the leaf $R$ rates revealed in this study can relate to an increased demand of intermediates and energy that can occur under stress [50–52]. Some acclimation adjustments in structural and functional changes in cell organization can be responsible for elevated demand, which is achieved through the increased respiration, although this increased $R$ rate was not associated with activation of the alternative respiratory partway in lettuce plants under the study. For all treatments under this study, leaf respiration in the light ($R_l$) was lower than in the darkness ($R_d$), which is consistent with numerous studies [53–58] and reflects the inhibition of the respiratory process by light. Previous studies have shown variability in $1- R_l{:}R_d$ depending on plant species, temperature, soil water content, nutrient availability, photoperiod, season [54,58–61], but this study did not reveal a significant effect of soil contamination with lead on the $R_l{:}R_d$ ratios (Table 1), reflecting no strong effect of Pb on light inhibition of foliar respiration in lettuce.

The $R{:}A$ ratio is an important indicator of the carbon balance of plants and ecosystems and can be useful in assessing the ability of plants to adapt to new growing conditions [62,63]. As a rule, the $R{:}A$ ratio tended to be constant under optimal growth conditions, but under conditions of stress, plants demonstrated an increased $R{:}A$ ratio [64]. This study did not reveal any differences between the $R_l{:}A_g$ and $R_d{:}A_g$ values in response to soil Pb contamination, although Atkin et al. [65] and Ayub et al. [66] showed higher stability of $R_l{:}A_g$ than $R_d{:}A_g$ to changing conditions. Elevated $R{:}A_g$ values in lettuce grown on the Pb-rich soil are associated with both the increase in leaf respiratory rate and the decrease in the net $CO_2$ assimilation rate and indicate a shift of the carbon balance towards greater carbon losses.

The results of this study showing that Pb accumulates in the roots to a greater extent than in the shoots confirms the findings that only a small part of absorbed Pb is transported to aerial parts of plants [5,7,15,16,67,68]. The root endoderm barrier can be responsible for the limited transport of Pb from roots to stems and leaves [10]. The localization of Pb in the insoluble fraction of root cell walls, as well as Pb sequestration in the

vacuole, has been shown to act as a detoxification mechanism [69]. Although, a high level of soil Pb limits the functional capacity of this barrier and Pb can be transported to the aerial parts of plants [7,67]. Farh et al. [8] reviewed the functional role of callose synthesized and deposited between the plasma membrane and the cell wall as a mechanical barrier against Pb stress. Moreover, Fe and Mn plaques can sequestrate Pb on surface roots to prevent its translocation within the root [70].

For lettuce plants tested in this study, soil contamination with lead mainly affected biomass allocation rather than total biomass accumulation, with higher biomass disposition in shoots than in roots. This study confirmed the results of earlier studies, which showed a greater accumulation of lead in the roots than in the aerial parts of plants, which is associated with the strategies that plants use to detoxify and survive under Pb stress [5,7,15,16,67,69,70]. Interestingly, while the concentration of lead in the shoots gradually increased during plant growth, the lead content in the roots reached a maximum in the early stages of growth and did not change thereafter. Moreover, the maximum concentration of lead in the roots varied between treatments and positively related to the content of soil Pb. The low effect of lead on plant biomass, found in this study, indicates that the plants were not strongly stressed at the low soil lead concentrations. Lettuce plants are likely to be able to tolerate low levels of lead in the soil by fixing lead ions in the roots without toxic effects on the plant and without accumulation of Pb to toxic levels dangerous to human health.

In summary, even though plants have used various strategies to tolerate contamination stress, including the accumulation of Pb in the roots and limitation of its transport to the shoots, depressed photosynthesis and stimulated respiration reflects an increase in carbon losses by plants, which negatively affect the carbon balance of plants under stress conditions of Pb contamination.

**Author Contributions:** Conceptualization, N.K.; methodology, E.I. and N.K.; investigation, E.I.; data curation, E.I.; writing—original draft preparation, E.I.; writing—review and editing, N.K.; visualization, E.I.; project administration, N.K. All authors have read and agreed to the published version of the manuscript.

**Funding:** This research was funded by the Russian Science Foundation, grant number 22-16-00145.

**Institutional Review Board Statement:** Not applicable.

**Informed Consent Statement:** Not applicable.

**Data Availability Statement:** Not applicable.

**Acknowledgments:** Experimental facilities for this study were offered by the Core Facility of the Karelian Research Centre of the Russian Academy of Sciences.

**Conflicts of Interest:** The authors declare no conflict of interest. The funders had no role in the design of the study; in the collection, analyses, or interpretation of data; in the writing of the manuscript; or in the decision to publish the results.

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
