# Peer review of "Physiological Responses of Lettuce (Lactuca sativa L.) to Soil Contamination with Pb"

_horticulturae, doi:10.3390/horticulturae8100951_

Round 1

Reviewer 1 Report

Authors target a good topic and data were collected nicely. The materials sections explained nicely. The manuscript can be accepted after the following revisions.

Please don’t start the sentences by abbreviations. For example, line 30, similarly improve throughout the manuscript.

Line 36, please confirm Ps or Pb

Why the following lettuce var. Medvezhje ushko was used in this study? Authors can write more information about it.

In materials and methods, authors should cite proper references by which the accomplished the experiments. For example, which methods or refences was followed during RWC measurement?

The discussion should be separated from the results, and should be write in precise form in comparison with previous literatures.

Conclusions should be rewrite and should be concise and precise.

Please avoid grammatical and typo mistakes.

Author Response

First we would like thank the Reviewers for attentive revision of our manuscript and constructive comments and suggestions.

Let me indicate the modifications made in the manuscript in the light of Reviewer’s comments.

Authors target a good topic and data were collected nicely. The materials sections explained nicely. The manuscript can be accepted after the following revisions.

Please don’t start the sentences by abbreviations. For example, line 30, similarly improve throughout the manuscript.

Thank You. Done.

We changed ‘Pb does not decompose and is not naturally removed from the soil [1]  to ‘It is known that Pb does not decompose and is not naturally removed from the soil [1];

‘Pb is not a plant nutrient, and a high Pb content negatively affects the morphological, physiological, and biochemical traits of plants, with the decreased crop production and yield [7–9]’ was changed to ‘Lead is not a plant nutrient, and a high Pb content negatively affects the morphological, physiological, and biochemical traits of plants, with the decreased crop production and yield [7–9]’;

‘Pb-mediated decline in photosynthetic rate was found to be a result of chlorophyll degradation…’ was changed to ‘The Pb-mediated decline in photosynthetic rate was found to be a result of chlorophyll degradation…’;

‘LMA values were used to calculate An at 1200 μmol PPFD m−2 s−1, Rl and Rd per unit leaf mass unit (An[mass], Rl[mass] and Rd [mass], respectively).’ was changed to ‘The LMA values were used to calculate An at 1200 μmol PPFD m−2 s−1, Rl and Rd per unit leaf mass unit (An[mass], Rl[mass] and Rd [mass], respectively)’;

Line 36, please confirm Ps or Pb

Thank You. We replaced Ps with Pb.

Why the following lettuce var. Medvezhje ushko was used in this study? Authors can write more information about it.

Yes, we added more information about lettuce variety under this study. We included ‘ This variety is popular in the Northwest of Russia because of this high productivity and good nutritional qualities’ colored with yellow in the text.

In materials and methods, authors should cite proper references by which the accomplished the experiments. For example, which methods or refences was followed during RWC measurement?

Thank You. The references are included in the section Materials and Methods:

‘The apparent quantum yield of photosynthesis (α) was calculated as the slope of the net CO2 assimilation rate (An[area]) versus irradiance of 20, 40 and 60 μmol m−2 s−1 according to Garmash and Golovko [34].’

‘The RWC value was calculated as (FW-DW)/(TW-DW)×100% according to González and González-Vilar [].’

The discussion should be separated from the results, and should be write in precise form in comparison with previous literatures.

Done. Thank You. We separated sections Results and Discussion.

Conclusions should be rewrite and should be concise and precise.

Now the conclusions are summarized at the end of the section Discussion.

Please avoid grammatical and typo mistakes.

Thank You. Done. All corrections are colored by yellow throughout the text.

Thanks one more to you!

Best regards,

Reviewer 2 Report

The paper titled „Physiological responses of lettuce (Lactuca sativa L.) to soil 2 contamination with Pb” is interesting however there are some questions that should be answered before paper publication.

Introduction

There is lack of information concerning testing lettuce (Lactuca sativa L.) in the context of Pb exposition. Authors should add a background in which important facts of lettuce response of lead should be presented.

Materials and methods

Authors stated that soil what was used for the experiments was not fertilized nor subjected to pesticide treatments for some years before soil collection.  In my opinion the soil have to be tested at least on the concentration of Pb and other heavy metals to prove that soil is not contaminated and all concentration of Pb comes from Pb(NO3)2 added by authors. Otherwise, all presented results are speculations.

There are also no information concerning homogenization and stabilization period of soil after Pb amendment. Was soil contaminated/supplemented by Pb(NO3)2 and used without stabilization period? What was the concentration of Pb in soil after its contamination and after 21, 34 and 46 days of experiment? Was all Pb bioavailable for lettuce?

Results and discussion

Figure 1 is unreadable – it should be corrected to be better readable

Authors made an attempt to explain the mechanisms, the effects of which they observed during the conducted experiments. In some places, the discussion contains information that is not referenced in the results. A serious mistake is the lack of reference of the obtained results (values) to the literature data. Authors should compare the values with the data presented in the literature by other scientists. In my opinion, this is a necessity.

Conclusion

This section should be rewritten due to mixed parts of discussion and conclusion. Some sentences should be placed (or were already placed) by authors in discussion section – lines 335-337, 348-350. Please make some corrections.

Author Response

First we would like thank the Reviewers for attentive revision of our manuscript and constructive comments and suggestions.

Let me indicate the modifications made in the manuscript in the light of Reviewer’s comments.

The paper titled „Physiological responses of lettuce (Lactuca sativa L.) to soil 2 contamination with Pb” is interesting however there are some questions that should be answered before paper publication.

Introduction

There is lack of information concerning testing lettuce (Lactuca sativa L.) in the context of Pb exposition. Authors should add a background in which important facts of lettuce response of lead should be presented.

Thank You. In the section Introduction, we added more information about the effect of Pb on the physiological traits of lettuce plants.

Line 72-86. ‘A decrease in the growth of roots and shoots, as well as an imbalance in nutrition and a change in the chlorophyll content was revealed for lettuce plants irrigated with lead-contaminated water for 15 days [7], but no significant effect of lead on root and shoot dry weight accumulation in lettuce plants was found by Đurđević et al. [] and Silva et al. []. The reason for these differences may be that the accumulation of Pb in lettuce is genotype-dependent, as was established by Zhang et al. []. Silva et al. [] showed significant DNA damage when lettuce crops were irrigated with water containing Pb, although seed germination parameters and seedling growth were not affected. In lettuce plants, an increased content of soil Pb reduced the contents of leaf macro- and micronutrients, as well as the content of photosynthetic pigments []. Pb-mediated depression of photosynthetic activity was found for lettuce plants [], but the response of respiration, and the coupling between photosynthesis and respiration to soil Pb contamination are not well documented.’

Materials and methods

Authors stated that soil what was used for the experiments was not fertilized nor subjected to pesticide treatments for some years before soil collection.  In my opinion the soil have to be tested at least on the concentration of Pb and other heavy metals to prove that soil is not contaminated and all concentration of Pb comes from Pb(NO3)2 added by authors. Otherwise, all presented results are speculations.

There are also no information concerning homogenization and stabilization period of soil after Pb amendment. Was soil contaminated/supplemented by Pb(NO3)2 and used without stabilization period? What was the concentration of Pb in soil after its contamination and after 21, 34 and 46 days of experiment? Was all Pb bioavailable for lettuce?

According to your comment, we added the information of the soil incubation period and the soil Pb content before and after the experiment period:

Line 95. ‘The available Pb content of soil under the study was 0.13±0.01 mg kg-1.’

Line 98-100. ‘Before seed sowing, the soil substrates of each treatment were incubated under 21–23 °C of air temperature and 70–80% of the maximum soil water holding capacity for 14 days, and then parked into plastic 0.80 L pots with a soil bulk density of approximately 1.4 g cm−3’.

Line 243-245. ‘At the end of the experiment, the concentration of available Pb in the soil was 0.10±0.02, 4.03±0.50, and 41.6±3.6 mg kg-1, respectively for 0Pb, 50Pb, and 250Pb treatment’.

Results and discussion

Figure 1 is unreadable – it should be corrected to be better readable

Thank You. Done. Figure 1 is changed.

Authors made an attempt to explain the mechanisms, the effects of which they observed during the conducted experiments. In some places, the discussion contains information that is not referenced in the results. A serious mistake is the lack of reference of the obtained results (values) to the literature data. Authors should compare the values with the data presented in the literature by other scientists. In my opinion, this is a necessity.

Thank You. We added we have added comparisons of our results with literature data (colored with yellow in the text):

‘Đurdecic et al. [] also did not reveal significant effect of Pb on leaf and root dry mass accumulation for lettuce plants’.

‘Our finding that photosynthetic activity depressed under soil Pb contamination is according to the studies have found that Pb can affect photosynthetic apparatus, including Calvin cycle, CO2 ability, photosynthetic pigment biosynthesis [7, 16, Khan 31, Mahaffey 32]’.

‘In our study, Pb soil contamination inhibited lettuce photosynthetic activity without any stimulatory effect of low Pb concentration on stomatal conductance, chlorophyll content, or net assimilation rate, as found in early studies [15–17].’

‘These results confirm the findings of Silva et al. [Silva, Pinto] that Pb-mediated limitations of photosynthesis occurred from reasons for reasons other than reduced CO2 availability’.

‘Previous studies have shown variability in 1- Rl:Rd depending on plant species, temperature, soil water content, nutrient availability, photoperiod, season [47, 51–54], but this study did not reveal significant effect of soil contamination with lead on the Rl/Rd ratios (Table), reflecting no strong effect of Pb on light inhibition of foliar respiration in lettuce.’

‘Ahmed et al. [] showed that total chlorophyll content of lettuce plants is sensitive to metals content in soil solution. The results of our study are consistent with the findings demonstrating decreased chlorophyll content in plants grown on soils contaminated with lead [Shu; (Seregin and Kozhevnikova, 2005]’.

‘Our results showed that leaf respiration is enhanced in lettuce grown in lead-contaminated soil, consistent with a number of studies showing a similar response of leaf and mitochondrial respiration to lead [18, 19].’

‘The results of this study showing that Pb accumulates in the roots to a greater extent than in the shoots confirms the findings that only a small part of absorbed Pb is transported to aerial parts of plants [5, 7, 15, 16, 60, 61].’

Conclusion

This section should be rewritten due to mixed parts of discussion and conclusion. Some sentences should be placed (or were already placed) by authors in discussion section – lines 335-337, 348-350. Please make some corrections.

The conclusion is rewritten and replaced in the section Discussion.

Thanks one more to you!

Best regards,

Round 2

Reviewer 2 Report

Paper was corrected according to reviewer suggestions. I accept paper in the present form.